# An Efficient Method to Determine Membrane Molecular Weight Cut-Off Using Fluorescent Silica Nanoparticles

**DOI:** 10.3390/membranes10100271

**Published:** 2020-10-01

**Authors:** Mariam Fadel, Yvan Wyart, Philippe Moulin

**Affiliations:** Aix Marseille Univ, CNRS, Centrale Marseille, M2P2 UMR 7340, Equipe Procédés Membranaires (EPM), Europôle de l’Arbois, BP80, Pavillon Laennec, Hall C, 13545 Aix en Provence CEDEX, France; mariamfadel.mf@gmail.com (M.F.); yvan.wyart@univ-amu.fr (Y.W.)

**Keywords:** molecular weight cut-off, membrane, nanoparticles, retention rate, particle tracking analysis

## Abstract

Membrane processes have revolutionized many industries because they are more energy and environmentally friendly than other separation techniques. This initial selection of the membrane for any application is based on its Molecular Weight Cut-Off (MWCO). However, there is a lack of a quantitative, liable, and rapid method to determine the MWCO of the membrane. In this study, a methodology to determine the MWCO, based on the retention of fluorescent silica nanoparticles (NPs), is presented. Optimized experimental conditions (Transmembrane pressure, filtration duration, suspension concentration, etc.) have been performed on different membranes MWCO. Filtrations with suspension of fluorescent NPs of different diameters 70, 100, 200 and 300 nm have been examined. The NPs sizes were selected to cover a wide range in order to study NPs diameters larger, close to, and smaller than the membrane pore size. A particle tracking analysis with a nanosight allows us to calculate the retention curves at all times. The retention rate curves were shifted over the filtration process at different times due to the fouling. The mechanism of fouling of the retained NPs explains the determined value of the MWCO. The reliability of this methodology, which presents a rapid quantitative way to determine the MWCO, is in good agreement with the value given by the manufacturer. In addition, this methodology gives access to the retention curve and makes it possible to determine the MWCO as a function of the desired retention rate.

## 1. Introduction

Membrane technologies are showing an increased importance in separation techniques, because they overcome many constraints linked to conventional processes. The use of these membrane processes is spreading in many sectors of industrial activities, notably in dairy, food and beverages, and in the manufacturing of chemicals, as well as in wastewater treatment, allowing for recovery and reuse of already wasted materials. This action allows these industries to become more environmentally friendly by decreasing the amount of waste generated, and also more profitable by the value components being recovered and reused. The main key for such revolution is the proper selection and application of these commercially produced and applied membranes [1,2,3]. The initial selection of the membrane for any application is generally based on the “Molecular Weight Cut-Off” or “MWCO” [2]. A membrane’s MWCO is defined as the minimum molecular weight of a solute that is 90% retained by the membrane, according to the French Standard NF X 45-103 [4]. The MWCO assessment of a membrane, facilitates theoretically the design processes of the engineering membrane systems and thus, ensures their predictable performance of retention [5]. There are a number of problems and limitations regarding the application and the use of MWCO [6,7]. Despite this, the measurement and comparison of MWCO curves and the determination of MWCO remain the most practical and universal means used to choose and differentiate between membranes for different applications, before evaluating a specific membrane in the real system of interest. Therefore, there is a need for accepted standards, leading to fast and inexpensive methods for MWCO determination. Thus, a robust, cost-effective, and fast procedure for the determination of MWCO is necessary for the current and future membrane industry, increasing applications and uses of membranes for the development of the membrane research area. This therefore implies a membrane retention study to be carried out in order to obtain a MWCO value of the chosen membrane. Currently, for ultrafiltration (UF) and microfiltration (MF) membranes, an acceptable, fast, and cost-effective universal standard method is still unavailable. In the methods currently available in the literature, a range of compounds with different molecular weights, such as polyethylene glycols (PEG) [5,8,9,10,11], oligostyrenes [12,13], alkanes [14,15], dextrans [8,11,16,17,18,19,20], pesticides [21], hormones [22], heavy metals [23], dyes from the textile industries [24] and acids [10], were used as the solute in the study of their membrane retention. Most of these tested methods are expensive and laborious. For example, four or five separate filtration runs are needed to finally obtain the MWCO using simple compound solutions. Few rapid, reliable, and relatively inexpensive chromatographic techniques have been addressed that are capable of separating a relatively cheap mixture of the different molecular weight compounds to allow a test of a single filtration to be used in the determination of MWCO [1,5,16,25]. However, there are limitations concerning the application of these techniques. For instance, the developed oligostyrene [13,26] and the protocols using PEGs [1] are for non-aqueous solvents. Furthermore, this old method requires the use of relatively expensive reagents and the precision and robustness of the latter, not yet well proven, in particular by benchmarking against the membranes available on the market. Thus, It is important to choose the mixture of solute to characterize the UF and MF membranes well, since blocking the pores of larger molecules on the surface of the membrane [5,18,27], or membrane fouling in forms of retention of large foulants [28,29], droplets deformability [30], and thermodynamics filtration resistance in gel/cake layer [30,31,32,33] can potentially selectively impede the transport of different molecular weight solutes, and therefore deflect the MWCO value from the real value [34]. In most cases, the major factor which does not allow validation of tests, with a single filtration of several components, is the possibility of concentration polarization affecting the membrane filtration [34]. Thus, it can result in denaturing the retention of solutes, and giving apparent retention which is specific to the conditions only to this system of filtration. This is the reason why continuous single-component filtrations to determine the MWCO are often chosen. [14,29]. Hermia proposed models that explain the main fouling mechanism (s) operating during filtration [35]. The classical blocking filtrations are described as complete blocking, standard blocking, intermediate blocking, and cake filtration, as schematized in Figure 1 [36]. The complete blocking mechanism is described as the foulant particle arriving at the membrane being bigger than the pore wall; thus, it entirely closes off the membrane pores. Standard blocking is described as the foulant particle being smaller than the pore wall, which leads to constriction on the membrane pores. Intermediate blocking is similar to complete blocking in that the pore is closed off at the membrane surface, but different in that the blocking is caused by more than one particle. Cake filtration refers to the buildup of foulant particles on the membrane surface due to complete blocking and intermediate blocking.

Several techniques have been used to identify the structure and characterize the foulants (tracers) in the membrane, such as, energy dispersive X-ray analysis [37], high performance size exclusion chromatography (HPSEC) [38,39], UV-absorbance and fluorescence spectroscopy [40,41], 3D fluorescence spectroscopy [41,42], scanning electron microscopy [38,42], and Fourier transform infrared spectroscopy (FTIR) [38]. Among all these techniques, fluorescence is increasingly used in the characterization of filtration membranes [43] because it makes it possible to target the desired information. It acts as a contrast agent which allows the direct, reliable and simplified identification of the compounds of interest. The use of fluorescence therefore makes it possible to obtain qualitative information on the morphology of a membrane through: the differences in density that it presents, the determination of surface roughness, and the size of the pores, whether or not followed by microscopic analysis of the membrane. The use of defined tracers also makes it possible to consider the effects of charges that can take place between the solutes and the membrane and can serve as a test of the integrity of the membranes. Fluorescent nanoparticles have many proven advantages: (1) they have dimensional stability which makes them non-deformable under pressure, (2) they do not, under certain conditions, aggregate between themselves, (3) they can be manufactured in large quantities and with precise sizes, and (4) they do not have preferential adsorption on the surface of the membrane, as in the case of organic or living compounds [44]. The information collected either by microscopy or by displacement techniques does not give a direct indication of the membrane selectivity that one would expect under normal conditions of use of the membrane. However, for the user, one of the main objectives of the characterization of the membranes remains their effectiveness for a selected separation. Measuring the retention rate of a solute for a porous membrane is one of the means available to facilitate the selection of a membrane for the separation of a given solute from a real fluid (NF-X45-103 1997) [4]. The principle consists of measuring the retention of a series of fluorescent NPs (tracers) of different sizes to obtain a selectivity curve as a function of the NP size. Numerous studies tried to find methods to determine the membrane pore size, but until today there has been no consistent method. A recent study, in 2020, used a mixture of gold nanoparticles solutes in a filtrate membrane to determine its pore size qualitatively, although the retention efficiency results are not consistent with 3 out of 5 tested membranes and many questions remain to be answered to understand this phenomenon fully, like why there is higher retention with small gold nanoparticles than with the larger particles [34].

Therefore, the objective of this work is to define an efficient method of measuring MWCO to provide information for membrane users to assist in membrane selection. To achieve this objective, the MWCO of ceramic membranes were evaluated by the retention and counting of fluorescent silica nanoparticles (NP) using particle tracking analysis. Description of the mechanism on NPs retention in the membrane is determined using both NPs in the feed and in the permeate which were collected versus time using tangential filtration.

## 2. Materials and Methods

### 2.1. Membranes

Mono channel ceramic membranes, with tubular shaped geometry, made of oxides or silicon carbide with high resistance mechanical materials, were used in this study. Membrane M1 and M2 present 0.1 and 0.2 µm molecular weight cut-off (MWCO) according to the manufacturer (Orelis, Salindres, France). The inner diameter of a channel is 6 mm and the external one is 10 mm. Filtration experiments were performed with membranes of 25 cm length. These membranes M1 and M2 have, respectively, an average permeability of 1620 and 3840 L·h^−1^·m^−2^·bar^−1^ with a standard deviation of 50 L·h^−1^·m^−2^·bar^−1^. Under filtration conditions, at neutral pH (7 ± 0.5), the membrane zeta potential is about −22 mV (constructor data). It is, therefore, of the same sign and intensity as the NP suspensions. This decreases the probability of NPs adsorption due to the repulsion forces between the NPs and membrane materials [45].

### 2.2. Nanoparticles and Analyses

With respect to the membrane pore size, the three NPs sizes used center on the MWCO of the membrane chosen. For M1- NPs 70, 100 and 200 nm were used and for M2-NPs 100, 200 and 300 nm. Spherical silica nanoparticles were used (Sicastar Red-F, Micromod Partikeltechnologie GmbH, Rostock, Germany) with diameters of 70 nm (NPs-70), 100 nm (NPs-100), 200 nm (NPs-200) and 300 nm (NPs-300). They are labeled with rhodamine, a fluorescent dye, covalently bound in the silica matrix. The rhodamine labeling does not affect NPs adsorption or retention. It has been reported that there is no variation in terms of flux and retention with addition of 50 mmol of NaCl in feed suspension compared to filtration without salt [45]. Nanoparticles have a hydrophilic surface with Si-OH end groups. Sizes were obtained with Zetasizer Nano S (Malvern, England) and with an individual monitoring particle analyzer NanoSight NS300 (Malvern, England) based on Nanoparticle Tracking Analysis (NTA). All measurements were performed at 25 °C. Using the Nanosight NS300, the average size and the size distribution for each NPs suspension were detected (Table 1). Four different NPs size suspensions were prepared with a high concentration. These high concentrations were selected to obtain a significant concentration in the permeate side, after filtration, to take into account the apparatus measurement range (between 10^7^ and 10^9^ part.mL^−1^). The pH of a solution can play a role in the chemical form of fluorophores. A change in pH therefore leads to a distorted quantification of the target species, so it is important to specify at which pH the measurements are taken. The pH of the suspensions is equal to 7. The feed concentrations of these suspensions are: 7 × 10^10^, 5 × 10^10^, 3 × 10^10^ and 9 × 10^9^ part.mL^−1^ for NP-70, NP-100, NP-200 and NP-300, respectively. For these high concentrations, a dilution is necessary before analysis. The main characteristics of NPs suspensions are summarized in Table 1. Samples were taken from each NPs feed solution and diluted before determining their size distribution using Zetasizer Nano S. The distribution curves of the four different NPs size suspensions are presented in Figure 2. The strength of these NPs used is their composition of solid and structured materials (silica) which make them incompressible and keep their shape under different operating conditions. The size of NPs will be the only reason why they pass through the membrane. Considering the zeta potential of NP suspensions (close to or greater than −30 mV) [45] and the fact that only NP are present in suspensions, the aggregation phenomenon is negligible and validated by microscopy observations and by Nanosight measurements.

### 2.3. Filtration Experiments

To remove the eventual residues before starting the filtration process, the membranes were flushed with Milli-Q water (conductivity of 0.8 µS·cm^−1^) under 1 bar of transmembrane pressure (TMP). Then, a measurement of permeability to ultra-pure (UP) water was carried out. M1 and M2 permeability was found equal to 1620 and 3840 L·h^−1^·m^−2^·bar^−1^, respectively. All filtration experiments were performed in vertical cross flow filtration mode using the lab-scale setup presented in Figure 3. The module containing the membrane is placed vertically in the filtration system (Figure 3), so that filtration is carried out as homogeneously as possible, avoiding preferential deposits. This vertical configuration is identical to that used for the industrial applications. Tangential filtration was carried out at constant flowrate of 90 L·h^−1^ and constant TMP of 0.5 bar ± 0.05 which was recorded (LEO Record, KELLER, Winterthour, Switzerland) versus time. The experiment was performed in an air-conditioned laboratory (20 °C ± 1 °C) but the temperature was followed throughout the experiment in order to correct the measured flux at 20 °C in agreement with the variation of water viscosity. Pure pressurized air was connected to the tank containing the NPs suspension, which was connected to the module. The feed solution is a suspension of NPs in Milli-Q water ultrafiltered by membrane (Pore size 20 nm, ALTEONTM I, Suez Aquasource^®^, Péone, France). During the filtration process, permeates are collected over time, almost every 6 to 10 s, and recorded by an electronic balance (Δm = ± 0.01 g) (Mark Bell, Berlin, Germany): the permeate flux was calculated. The permeate is collected continuously but by a volume of approximately 5 mL which is the minimum volume needed to make a nanosight analysis. At the end of each filtration, the membrane was disconnected from the module and a backwash was performed at 3 bars with a 4 L solution of Citric Acid (68%, Fisher Scientific, Illkrich, France) of pH ~ 1.8, heated at 50 °C to destroy/remove the NP and the membrane fouling. Then, the membrane was rinsed with Milli-Q water at 3 bars. If the membrane permeability was not recovered, a new acid backwash was performed. Each filtration experiment (one membrane – one NP size) was repeated at least 2 times, to ensure the reproducibility of the results.

### 2.4. Nanoparticle Tracking Analysis Techniques—Nanosight (300)

Permeates at various times, and feed and retentate at final filtration time were analyzed by Nanosight NS300 “NTA” (Malvern, England). This technique is based on NTA (Nanoparticle Tracking Analysis), and the identification of the Brownian movement of nanoparticles makes it possible to go back to the size of the latter. A fluorescence mode allows clear and precise identification of the nanoparticles used. This device was therefore used to obtain, in addition to the concentration of the flows, the size distribution of the suspensions of nanoparticles of different sizes chosen. To minimize the fluorescence fading of NPs during the filtration runs, the permeate collection beaker was wrapped with aluminum foil. The optimal measurement range of NTA is from 10^7^ to 10^9^ part.mL^−1^; suspensions were diluted for analysis purposes when necessary (feed and retentate essentially). The aggregation, which could be the cause of a greater retention, was controlled by comparing NPs size in feed and retentate. The results validate that no aggregation occurred. Each sample measurement consisted of 3 scans, and was performed under a calibrated constant continuous flow from a syringe of 1 mL. The flow to be applied must be strong enough to increase the number of particles counted (in comparison with a static measurement where only the particles which are present in the analysis field are counted), but not too fast, otherwise the Brownian movement cannot be characterized for long enough (the particles pass through the fields, but do not have time to be followed). The particles must therefore take around 10 s to cross the screen, so the flow must be adjusted (the number to enter in the software) to obtain this speed.

The measurements quantitatively present the average concentration, means (defined as the average size of NPs), and mode (the most often value of NPs size) with their relative errors. These measurements were repeated at least 2 times to ensure their reproducibility. Importantly, the particle-tracking technique employed here provides direct evidence of fouling mechanisms under specific operating conditions. This valuable information could benefit the design and optimization of the filtration processes [46].

## 3. Results and Discussion

### 3.1. Retention Rate

#### 3.1.1. Retention of NPs

After preparing the feed solution, the first step is to validate and optimize the parameters of the Nanosight N300 analysis technique. Figure 4 presents the concentration distribution as a function of the NPs size of the prepared feed solution of 5 × 10^10^ of NPs 100 nm. The Nanosight measurements show 5.02 × 10^10^ part.mL^−1^ of average concentration, the average size of the NPs is equal to 106.1 +/− 0.2 nm, with a mode at 97.6 +/− 1.7 nm, which is compatible with the information given by the supplier.

Initial and final retention rates are calculated from the concentrations of the first permeate and the feed, and the last permeate and retentate collected, and analyzed at the end of the filtration, respectively. Under the same optimized detection parameters of Nanosight, the measurements of the other permeate samples were performed. Figure 5A shows the concentration distribution for each NPs size of the feed and permeate solutions after 30 min of the filtration of NPs 100 nm suspension with a membrane M2. It is obvious that the smallest NPs are completely filtered by the membrane as they are shown in the permeate curve. With the increase in the size of NPs, the concentration difference between the permeate and the feed increases. The retention rate [17,22,29] was calculated by the Equation (1), according to the French Standard NF X 45-103 [4] and the variation of the retention rate versus Np size is given in Figure 5B.
(1)$$Retention\;Rate_{t}\left(\%\right)=\left(1-\left(\frac{{\left[permeate\right]}_{t}}{{\left[Feed\right]}_{t}}\right)\right)\times 100$$
where [permeate]_t_ is the solute concentration in the permeate and [Feed]_t_ is the solute concentration in the feed at the same time t. The unit of concentration is in part.mL^−1^. With this new methodology, the retention curve can be drawn for each time, and the effect of membrane fouling on the retention rate can be estimated.

#### 3.1.2. Effect of Fouling on the Retention Rate

Filtration time showed an impact on the retention rate curves, although the curves had the same shape (Figure 6). The impact of the filtration time on the retention rate was determined by having filtration last 30 min and collecting the permeates at different times during the filtration process. MWCO is equivalent for 2 and 5 min, around 80 nm for a retention rate of 90%, but there is a stronger shift at 30 min. For this time, the MWCO is estimated around 60 nm for a retention rate of 90%. For M2-NPs100, the retention and accumulation of NPs-100 in/on the membrane during the passage of the first mL of permeate led to an increase in membrane fouling and a modification of selectivity. Therefore, the filtration time was optimized for 2 min to determine the selectivity of the membrane. This is compatible with the duration used in the previous methods which were based on a short time filtration experiment for the determination of the membrane pore size [34]. This time (2 min) is a compromise between having a sufficient volume of permeate for Nanosight analysis and having a minimum of fouling, in order to limit its impact on the determination of the MWCO.

#### 3.1.3. Influence of NPs Size on Retention Rate

As shown in previous studies [45], the size of NPs has an impact on their retention by the membrane. Since there is an interest in the selectivity of the membrane, the impact of NPs size of larger, close to, and smaller than the membrane pore, was tested. With the two membranes M1 and M2, the retention rate curves plotted against the size of NPs decreased (Figure 7). For the membrane M1, the retention of NPs presented a MWCO at 58.5–79.5 and 110 nm with NPs70-100-200, respectively. In general, small particles penetrate deeper and faster into the membrane [28,29]. Thus, for each NP size tested, the determination of MWCO is affected by the smallest NPs which are blocked in the membrane. So we can deduce that the larger the chosen size, the closer we get to a satisfactory determination of the MWCO as evidenced by the agreement with the manufacturer’s data and a study in the literature [34]. The same conclusion is obtained for the membrane M2, where the MWCO of the membrane was estimated at 85, 121.5 and 181.5 nm from the retention curves of NPs100-200-300, respectively. This small difference between the value determined by this method and the value quoted by the manufacturers can be addressed by the differences in methodology and test conditions [47]. Following the shape of the retention curves (Figure 7) within the different experiments performed, they show that they follow the same path with that of the increase in the size particle distribution of NPs. These results show the consistency of this method performed in two different membranes and each with three different sizes of NPs feed solutions.

To ensure the robustness of this methodology and thus the reproducibility of the results, the experiments were repeated at the same operating conditions. Figure 8 represents the retention rate of the NPs100 suspension filtrated with membrane M2 after 2min of the filtration process. The two curves of retention rate from the repeated experiments are almost identical, where the difference of the MWCO value is ± 0.6% between the two performed experiments.

### 3.2. Fouling Mechanism and MWCO Membrane

For each filtration run, the experimental data were plotted and compared to the linear forms of four fouling models (cake filtration, intermediate blocking, standard blocking and complete blocking) by calculating the correlation coefficients (R^2^) values [23,29]. It was decided to target relatively high R^2^ values (up to 0.95) in order to identify the most appropriate fouling model. Results showed that the permeate flux decreased during the filtration with all NPs size, mainly due to cake filtration fouling model (R^2^ = 0.99 in most of the cases) as shown in (Figure 9). Whatever the size of the NPs and the membrane molecular weight cutoff tested, at the end of filtration, the cake filtration appears. This shows very clearly that the determination of the molecular weight cut-off is affected by the duration of the filtration and the quantity of NPs retained. When the size of the NPs used is very small, in the first stages of filtration a standard blocking appears logically with penetration of the NPs into the membrane and then a cake filtration is formed. Conversely during the filtration of the largest NP sizes, a complete blocking then a cake filtration appears from the first minutes of filtration [46].

## 4. Conclusions

In this feasibility study, a new and fast method is presented to determine the molecular weight cut off (MWCO) of a membrane based on the retention of fluorescent NPs. Two Membranes—M1 and M2—of different pore sizes have been investigated. The reproducibility of the experiments was validated. The retention rates of NPs with sizes larger, closer to, and smaller than the membrane pore size given by the manufacturer at optimized experimental conditions were studied. It was shown that the retention of NPs changed over the filtration process due to the membrane fouling: as expected, a shift of the MWCO value is observed whatever the experiment after a certain duration of the filtration process. The membrane fouling was studied as a function of the NP size with the Hermia model. From a standard to a cake fouling modelling, the results are in agreement with the NP size. So, the selectivity of the membrane should be detected at the first few minutes of the filtration process. The membranes used presented good retention rates curves with different sizes of NPs, increasing gradually until they formed a plateau at 100% of retention. This allows determining the MWCO at a 90% retention rate according to the French Standard NF X 45-103 although another could be selected. A MWCO of 181.5 nm has been determined for Membrane M2 with NPs of 300 nm, and 110 nm for the Membrane M1 with the NPs of 200 nm, close to the values given by the supplier (200 and 100 nm, respectively). With this new methodology, it is possible to obtain a continuum of information of the retention rates versus the NP sizes, and as a function of the membrane application it is possible to obtain the retention rate for a specific molecule. These NPs can later be destroyed with an acid wash, to recover the permeability of the membrane. The robustness of this methodology is in progress and will be the subject of a forthcoming paper.

## Figures and Tables

**Figure 1 membranes-10-00271-f001:**
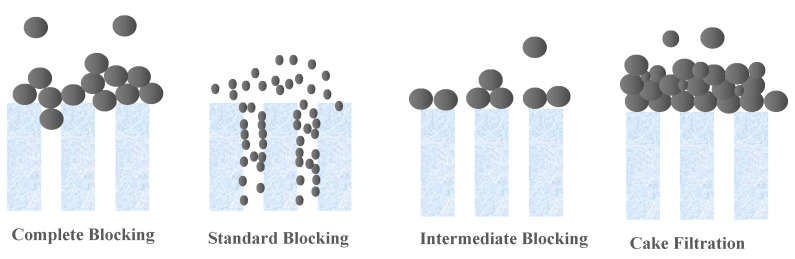
Schematic representation of various fouling mechanisms by particulate foulants [36].

**Figure 2 membranes-10-00271-f002:**
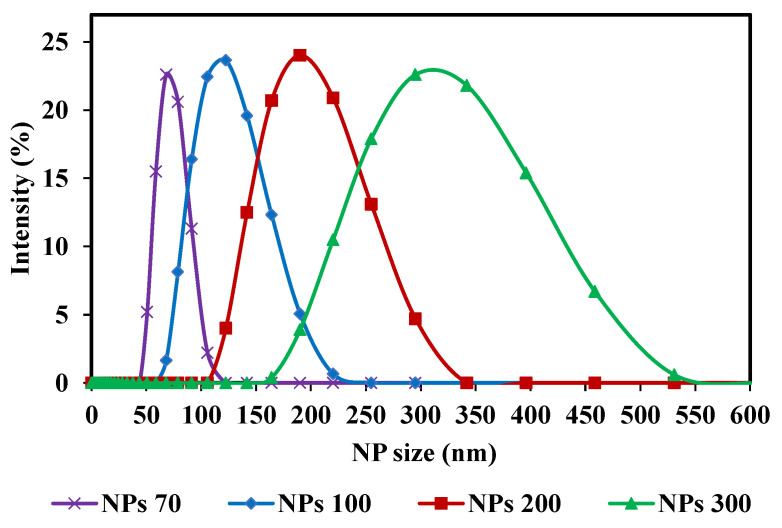
Nanoparticle size distribution curves (NPs) of the initial solutions (feed): NPs 70; 100; 200 and 300 nm, used in the filtrations.

**Figure 3 membranes-10-00271-f003:**
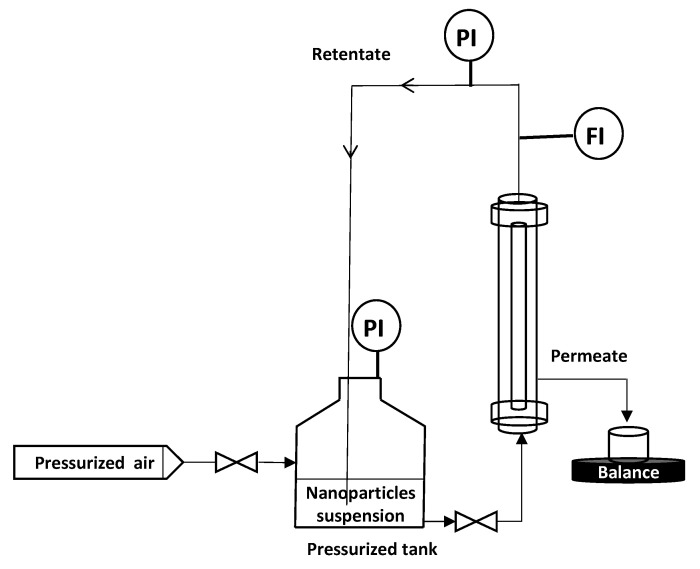
Schematic diagram of the vertical cross flow filtration set-up.

**Figure 4 membranes-10-00271-f004:**
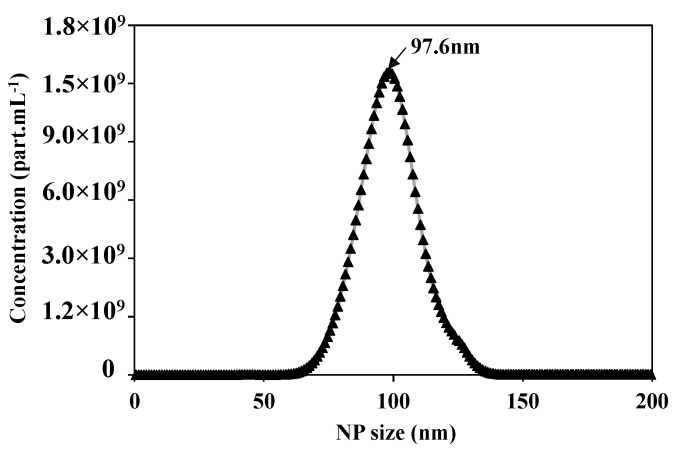
Concentration distribution for each NP size of suspension of NPs100 nm using the Nanosight N300 analytical technique.

**Figure 5 membranes-10-00271-f005:**
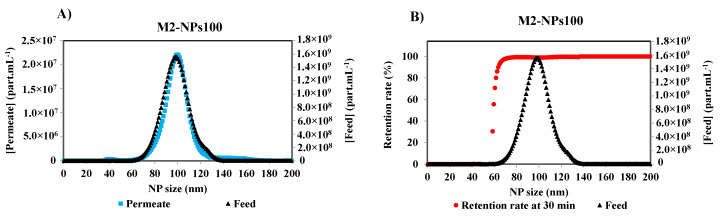
(**A**) Particle size distribution of the feed and the permeate of NPs100 at 30 min of the filtration process with membrane M2, and (**B**) associated retention rate at 30 min of filtration of M2-NPs100 [Room temperature, TMP ~0.5 bar].

**Figure 6 membranes-10-00271-f006:**
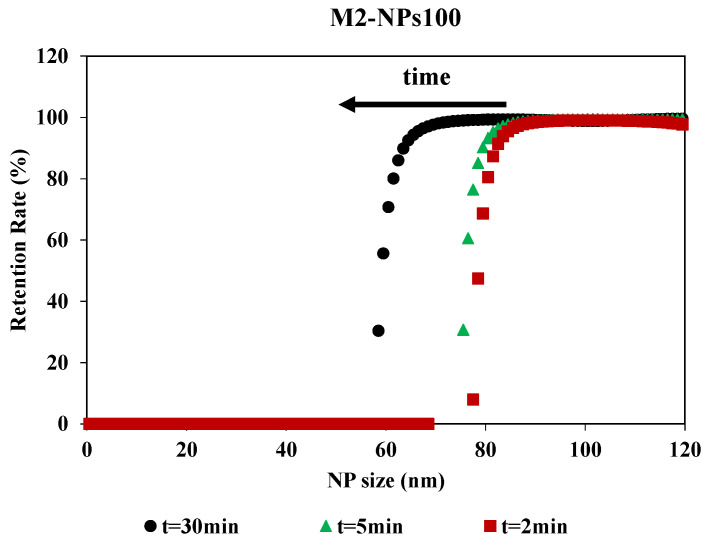
Retention rate of NPs100 in function of NPs size for the filtration with membrane M2 over the time realized [TMP 0.5 bar].

**Figure 7 membranes-10-00271-f007:**
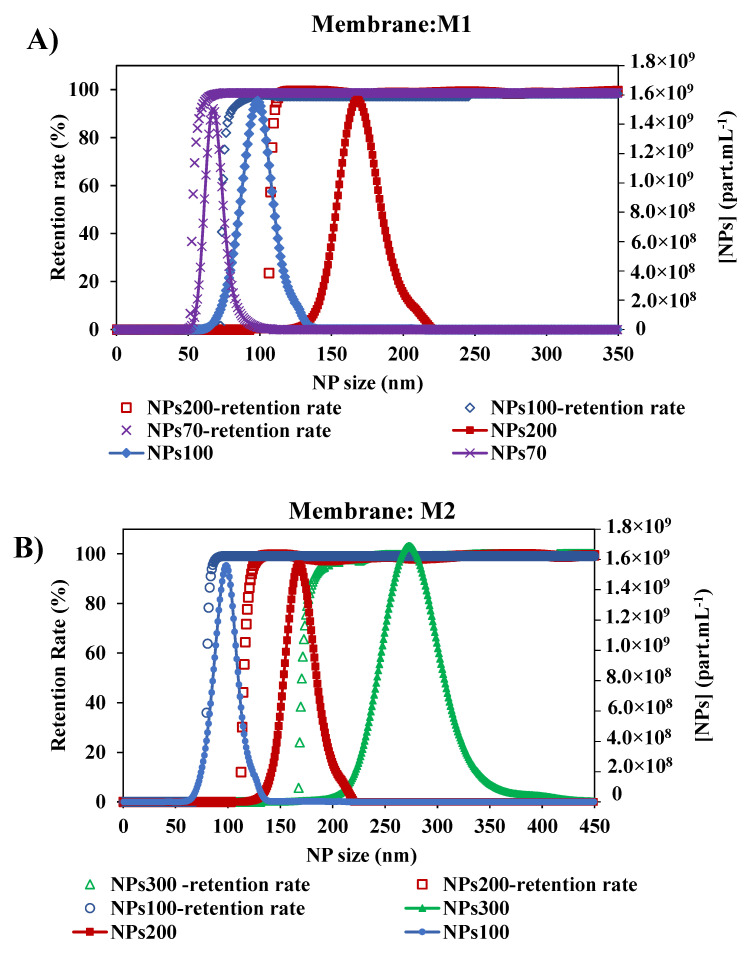
Retention rates of the different size of NPs with two membranes: (**A**) Membrane M1 with the initial suspension solution of NPs 200, NPs 100 and NPs 70 nm, and (**B**) Membrane M2 with the initial suspension solution of NPs 300, NPs200 and NPs100 nm. [TMP ~ 0.5 bar, room temperature].

**Figure 8 membranes-10-00271-f008:**
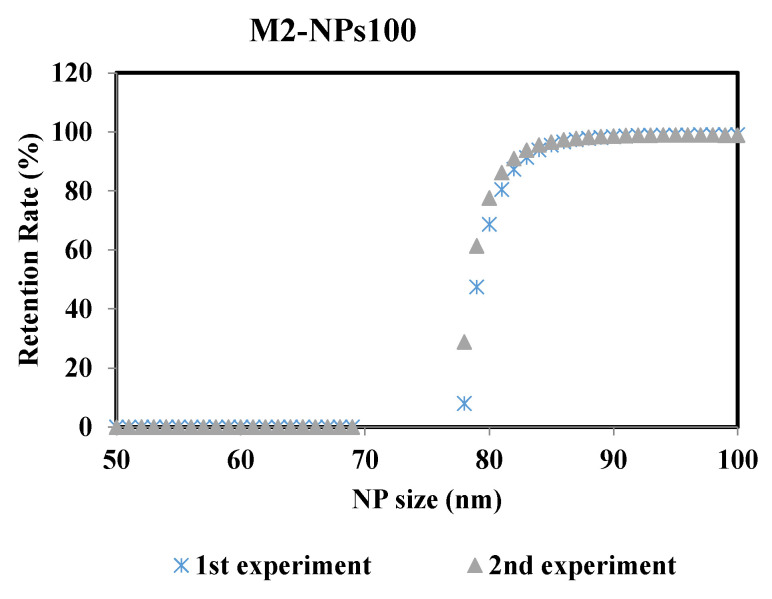
Retention rates of NPs 100 filtrated with membrane M2 of the two performed experiments: 1st and 2nd experiments carried out at same operating conditions.

**Figure 9 membranes-10-00271-f009:**
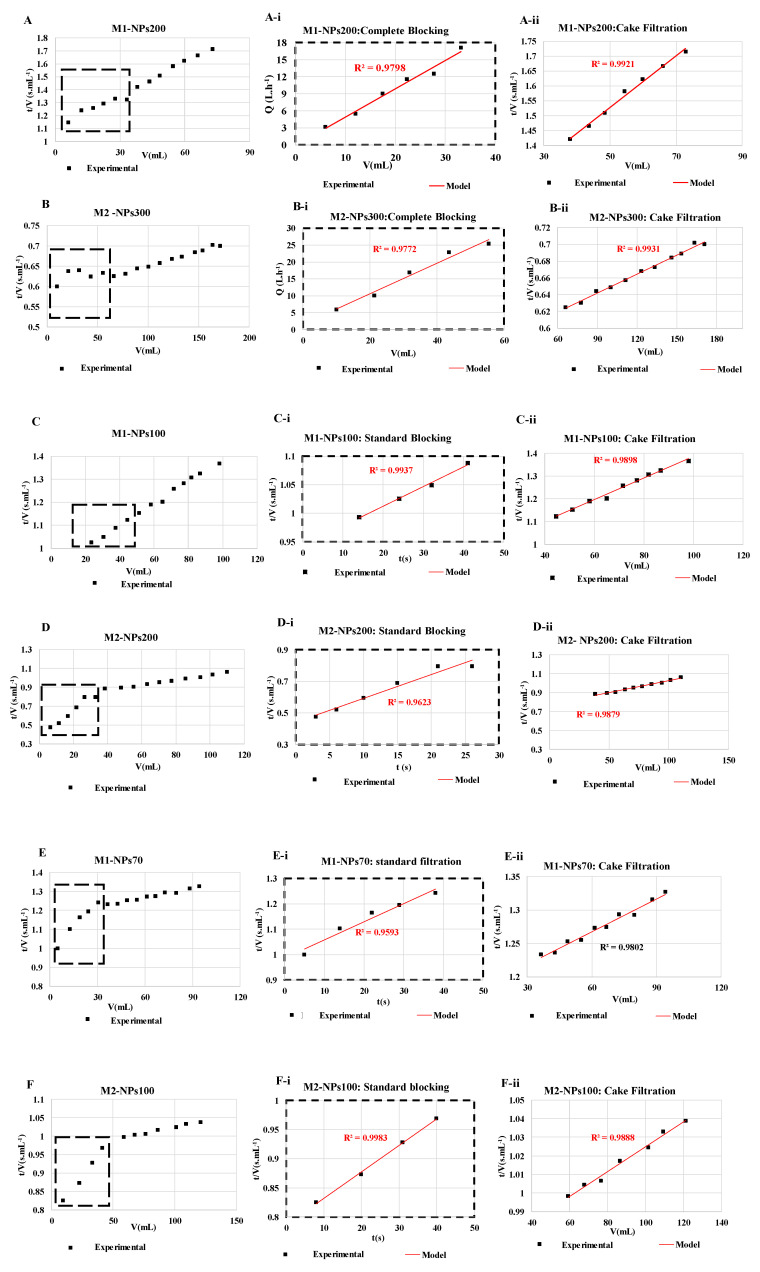
Fouling models found for NPs filtrations of (**A**) NPs 200 with membrane M1, (**B**) NPs 300 nm with membrane M2, (**C**) NPs 100 with membrane M1, (**D**) NPs 200 nm with membrane M2, (**E**) NPs 70 with membrane M1, and (**F**) NPs100 with membrane M2. The experiments are performed at same experimental conditions [TMP ~ 0.5 bar, room temperature].

**Table 1 membranes-10-00271-t001:** Physicochemical characteristics of the nanoparticles (NPs) used in suspensions.

Parameter	NPs-70	NPs-100	NPs-200	NPs-300
Diameter ^a^ (nm)	78.5	106.1	173.6	293.8
Mode ^a^ (nm)	66.3	99.9	164.9	288.8
Wavelength ^b^ (nm)	585	585	585	585
Feed concentration (part.mL^−1^)	7 × 10^10^	5 × 10^10^	3 × 10^10^	9 × 10^9^

^a^ Size average and mode values measured by Nanosight NS300. ^b^ Detection ranges of emission wavelengths given by manufacturer.

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
