# Peer review of "An Efficient Method to Determine Membrane Molecular Weight Cut-Off Using Fluorescent Silica Nanoparticles"

_membranes, 2020, doi:10.3390/membranes10100271_

Round 1
Reviewer 1 Report
The article reports the method of molecular weight cut-off determination using fluorescent silica nanoparticles. Some points have to be corrected and clarified in order to have a more understandable manuscript.
- Abstract.
- The authors are suggested to make it more concise. It should not contain the conclusions...
- Materials and Methods
- The authors are suggested to add the Materials section with the description of all the chemicals/materials they used.
- 2.1 Membranes and 2.3 Filtration section - Please check the permeability of M1 and M2. They are vice versa in these sections..
- 2.2 Nanoparticles and analyses - What is the methodology of the labeling process? It need to be added and/or the reference you follow should be mentioned. Was the labeling confirmed? If yes, how? Additional results that confirm the labeling are suggested.
- 2.3 Filtration experiments - "TMP" was not mentioned previously in the manuscript. The authors are suggested to check if they provide the full name of the abbreviations they used in the manuscript.
- 2.5 Fouling Models - Would it be better to move to the Introduction? This section contains only fouling mechanisms without any methods...
- Results and discussion
- Line 238 - Reference source not founded. Please check it.
- Why authors used only one membrane with one particle size in sections 3.1.1. and 3.1.2? No comments on other membrane and particles sizes were made. Additional table (or other representation of the data) will help.
- The authors are suggested to improve the entire section (results and discussion). It is hard to follow the authors in several places...
- Figure 9 - Additional information is needed. What is A, B, C, D, E, F? The authors are suggested to add the labeling information under the figure.
Author Response
see attached filed

Reviewer 2 Report
I found the paper very interesting. The design of the research and clarity in explanation is satisfying. The methodology and the importance of the work are well stated and discussed. My final decision is accept as is.
Author Response
I found the paper very interesting. The design of the research and clarity in explanation is satisfying. The methodology and the importance of the work are well stated and discussed. My final decision is accept as is.
Thank you very much for this report.
Reviewer 3 Report
The authors propose the use of fluorescent nanoparticles to determine the MWCO of commercial membranes. I consider this work could useful if published in a journal focused on methods and protocols, but I think Membranes is not a adequate journal for it. Only the results about fouling could be considered interesting from this point of view, but the work in this topic is scarce. They authors mention that "The robustness of this methodology is in progress and will be the subject of a forthcoming paper". I suggest this manuscript can serve as introduction to this new paper and more attention to fouling should be paid.
Author Response
Reviewer 3
The authors propose the use of fluorescent nanoparticles to determine the MWCO of commercial membranes. I consider this work could useful if published in a journal focused on methods and protocols, but I think Membranes is not a adequate journal for it. Only the results about fouling could be considered interesting from this point of view, but the work in this topic is scarce. They authors mention that "The robustness of this methodology is in progress and will be the subject of a forthcoming paper". I suggest this manuscript can serve as introduction to this new paper and more attention to fouling should be paid.
Thank you very much for your comment. We are agree with you: this paper is a feasibility to demonstrate the potential of this new method.
Round 2
Reviewer 3 Report
The changes have not modified my initial opinion: the manuscript should not be acepted for publication.
As I expresed in my first review report, I consider this work could useful if published in a journal focused on methods and protocols, but I think Membranes is not a adequate journal for it. Only the results about fouling could be considered interesting from this point of view, but the work in this topic is scarce. They authors mention that "The robustness of this methodology is in progress and will be the subject of a forthcoming paper". I suggest this manuscript can serve as introduction to this new paper and more attention to fouling should be paid.